

# Evaluating the oestrogenic activities of aqueous root extract of *Asparagus africanus* Lam in female Sprague-Dawley rats and its phytochemical screening using Gas Chromatography-Mass Spectrometry (GC/MS)

Abubakar El-Ishaq[1,2,*], Mohammed A. Alshawsh[3,*] and Zamri Bin Chik[2]

[1] Science Laboratory Technology Department, School of Science and Technology, Federal Polytechnic, Damaturu, Yobe, Nigeria
[2] University of Malaya Bioequivalence Testing Centre (UBAT), Department of Pharmacology, Faculty of Medicine, University of Malaya, Kuala Lumpur, Selangor, Malaysia
[3] Department of Pharmacology, Faculty of Medicine, University of Malaya, Kuala Lumpur, Selangor, Malaysia
[*] These authors contributed equally to this work.

## ABSTRACT

*Asparagus africanus* Lam. is a plant used traditionally for natal care. This study evaluates the oestrogenic activities of aqueous root extract and screens for possible bioactive phytochemicals. Oestrogenicity of *A. africanus* was evaluated in ovariectomised rats treated with 50, 200, and 800 mg/kgBW doses twice daily for three days. Ethinyl estradiol (EE)1 mg/kg was used as positive control, and hormonal analysis and gene expression were carried out. The findings demonstrated that the extract produced a dose-dependent increase in the oestrogen levels with a significant increase compared to untreated rats. Pre-treatment with oestrogen receptor antagonist (ORA) prior to *A. africanus* treatment reversed the trend. Gene expression analysis on rats treated with 200 mg/kgBW *A. africanus* showed significant ($p < 0.005$) upregulation of oestrogen receptor alpha (ERα), while pre-treating animals with (ORA) significantly ($p < 0.005$) increased the expression of calbindin 3 (Calb3) in the EE group as compared to the untreated rats. The GC/MS results showed the presence of steroidal saponins such as stigmasterol and sarsasapogenin. These might be the bioactive constituents that exhibited these activities. The oestrogenic properties of *A. africanus* revealed in this study could contribute to the antifertility properties of the plant. However, further pharmacological studies are required to confirm the antifertility effect.

Corresponding author
Zamri Bin Chik,
zamrichik@ummc.edu.my

## INTRODUCTION

According to the World Health Organisation (WHO), approximately 80% of the world's population relies on traditional medicine which involves the use of plant extracts (*Wachtel-Galor & Benzie, 2011*; *WHO, 2013*). This practice is more common among villagers where modern drugs are not available or are too expensive (*Adamu et al., 2005*). *Asparagus africanus* Lam is a plant that is used in traditional medicine for contraception and to assist women during parturition. In addition, the aqueous extract of the plant is believed to have cleansing properties, especially after parturition and its roots contain polyphenols, phytosterol, saponins, and tannins (*Yared, Mekonnen & Debella, 2012*). Other species of *Asparagus* (*Asparagus pubescens*) reportedly have the ability to reduce the number of pups output in rats, mice and rabbit species (*Nwafor, Okwuasaba & Onoruvwe, 1998*). Medicinal plants have various actions on a body's physiology, and some plants contain phytoestrogen and may cause fertility, whereas others have abortifacient and antifertility activities (*Mukta & Nagendra, 2015*).

Oestrogen is a vital hormone during development and maintenance of normal sexual and reproductive functions. Oestrogen is well known to be a morphogen and plays a vital role during morphogenesis of the uterus (*Heldring et al., 2007*). Oestrogen signalling pathways are selectively promoted or inhibited depending on the balance between the activities of oestrogen receptors (ERα and ERβ) in target organs. Oestrogen receptors (ERs) belong to the steroidal hormone superfamily of nuclear receptors, which act as transcription factors after binding to oestrogen (*Pillai, Jones & Koos, 2002*). In addition, oestrogen production is vital for proper implantation of the blastocyst with the uterine wall during pregnancy (*Jeff et al., 2001*). Oestrogens and progestins are established modulators of the reproductive function in normal cycling and remodelling during menses and pregnancy (*Crabtree et al., 2006*).

An uncontrolled population affects the socio-economic development of a country (*Crabtree et al., 2006*). To control population growth, there is a need for an acceptable female contraceptive (*Londonkar & Nayaka, 2013*). Family planning has been promoted through several methods of synthetic contraceptives, but synthetic drugs have many adverse effects. Hence, there is a need to search for safe medicinal plants with contraceptive potentials free from adverse effects (*Crabtree et al., 2006*). However, due to the lack of intensive and comprehensive investigation, no safe and effective oestrogenic plant has been found (*Londonkar & Nayaka, 2013*).

A model used to evaluate the effect of chemicals on estrogen receptors is the uterotrophic assay in which removal of the ovaries reduces endogenous estrogens, causing the uterus to shrink. Replacement of these hormones with external sources of estrogen causes a trophic response of the uterus (*Hye-Rim, Tae-Hee & Kyung-Chul, 2012*). The current study uses the uterotrophic assay to investigate the oestrogenic activities of aqueous extract of *Asparagus africanus* (AEAA) in female Sprague-Dawley rats to screen its phytochemical constituents using GC/MS.

## MATERIALS AND METHODS

### Chemicals and drugs

Meloxicam was obtained from Bal Pharma, India. Fulvestrant (ICI 182, 870) and 17α-ethinyl estradiol were obtained from Sigma-Aldrich, Inc. St Louis. RNase plus mini, and RNase-Free DNase kits were purchased from Qiagen, Germany. All chemicals are of analytical grade.

### Plant material collection and extraction

The sample of the *A. Africanus* plant was collected in Damaturu metropolitan, Yobe State, Nigeria. The plant was authenticated by plant Taxonomist (Dr. Kien-Thai, Yong) and the sampled plant was assigned a reference number (KLU 48696) and deposited in the Herbarium, University of Malaya. The root plant part was processed into a fine powder under laboratory conditions, and 50 g of fine powder was extracted in 500 mL of distilled water for 24 hr (*Dunn, Turnbull & Sharon, 2004*). The extract was filtered using Whatman filter paper No.4. and the filtrate was kept in the freezer at −20 °C before being evaporated to dryness in a vacuum using freeze dryer (Eyela FDU-1200; Tokyo, Japan) maintained at −50 °C for three days. The obtained yield of crude extract (7.0%) was kept in a fridge at 4 °C for further analyses.

### Evaluation of the oestrogenic activity of *A. africanus*

All experimental procedures were carried out according to the approval of the ethics committee for animal experimentation, Faculty of Medicine, University of Malaya, Malaysia (Ethics Reference no.:2015-180505/PHAR/AEI). The female Sprague-Dawley rats were obtained from the animal experimentation unit, Faculty of Medicine, University of Malaya. The animals were housed in polycarbonate cages in a controlled environment (temperature, 23 °C ± 2 °C; relative humidity, 50 ± 10%; frequent ventilation; and an illumination schedule of 12-h light/12-h dark). The experimental animals were allowed free access to phytoestrogen-free diet (Altromin 1324 FORTI) and tap water. Thirty immature bilaterally ovariectomised female Sprague-Dawley rats with body weights ranging from 160–180 g were used for this study. The rats were randomly divided into five groups comprising six each ($n = 6$). Three groups of animals were administered a subcutaneous (SC) injection of *A. africanus* extract dissolved in distilled water ($DH_2O$) at three different doses (50, 200 and 800 mg/kgBW) twice daily for a total doses of 100, 400, and 1,600 mg/kg per day for three consecutive days from day 14 to day 16 post ovariectomy (*Bo-Mi et al., 2009*). Corn oil (5 ml/kg BW) and 17α-Ethinyl estradiol (EE: 1 mg/kg BW) were administered SC and served as negative and positive control, respectively. The animals were monitored for any clinical signs and abnormal behaviours daily throughout the experimental period (*Jain et al., 2016*).

In the second batch of the experiment, another 30 rats ($n = 6$) were pre-treated with SC injection of 1 mg/kg BW fulvestrant (ICI 182, 870), which is an oestrogen receptor antagonist (ORA), 30 min before each treatments. All animals were observed for vaginal cornification and early vaginal opening (*Marcondes, Bianchi & Tanno, 2002*; *Srivastava et al., 2007*) and were euthanised 24 h after the last dose of treatment using an intraperitoneal

overdose of 150 mg/kg ketamine and 15 mg/kg xylazine. The body weights of the animals were recorded, and the uteri tissues were harvested for gene expression analyses (*Bo-Mi et al., 2009*; *Deepak, Rema & Roy, 2015*). The blood was collected, and the separated serum was used for hormonal analysis (*Yoshinaka et al., 2009*).

## Hormonal assay

ELISA kits (Thermo Fisher Scientific, Waltham, MA, USA) were used to measure oestrogen, progesterone and luteinising hormone (LH) levels. Briefly, the plate was washed manually with 350 µl washing buffer and soaked for two minutes, followed by aspiration of the content from the plate. The process was repeated three times. 50 µl of standard or sample was added to each well and 50 µl of antibody was added immediately to each well and incubated for 45 min at 37 °C. The plate was washed three times and 100 µl HRP-Streptavidin Conjugate (SABC) working solution was added to each well and further incubated for 30 min at 37 °C. Aspiration was continued, and washing procedures were repeated five times. Then, 90 µl of tetramethylbenzidine (TMB) substrate solution was added and the content incubated for 15–20 min at 37 °C. Finally, 50 µl of stop solution was added, and the absorbance was measured at 450 nm immediately. The result was extrapolated from the plotted standard curve (*US Food and Drug Administration, FDA, 2016*).

## Gene expression assay

Quantitative reverse transcription polymerase chain reaction (RT-PCR) was carried out to identify the expression levels of oestrogen receptor alpha (ERα) and calbindin-D9K (Calb3) genes in treated and untreated rats. Total RNA was extracted from uterus tissues using RNase plus mini kit (Qiagen, Hilden, Germany), according to the manufacturer's protocol.

The concentration and purity of extracted RNA samples were determined by NanoDrop spectrophotometer 2000c (Thermo Fisher Scientific, Waltham, MA, USA), while the quality of RNA was further determined by Bioanalyzer Agilent RNA 6000 Nano Kit (Agilent, USA). One microgram of total RNA was reverse transcribed to cDNA using high capacity RNA-to cDNA Master Mix supplied by Applied Biosystems (Foster City, CA, USA). The obtained cDNA was stored at −20 °C until further used. For RT-PCR, ERα (Rn01640372_m1) and calbindin (Rn00560940_m1) assay genes were supplied from TaqMan (Applied Biosystems). A volume of 1 µl complementary cDNA was used for qPCR which was performed using standard conditions. The amplification and quantification reactions were performed using the StepOne Plus Real-Time PCR System (Applied Biosystems, USA) according to TaqMan gene expression assay protocol. GAPDH (Rn0177563_g1) and HPRT-1 (Rn01527840_m1) were used as the endogenous reference genes. Cycling kinetics was performed in 40 cycles, to ensure linearity of PCR product (*Thuy & Eui-Bae, 2009*). The real-time PCR reaction was performed in triplicate, and the averages of the obtained threshold cycle ($C_t$) values were processed for further calculations according to the comparative $C_t$ method.

Gene expression values were calculated according to the $2^{-\Delta\Delta C_t}$ method (*Livak & Schmittgen, 2001*). The $\Delta C_t$ value of each sample was determined by subtracting the

average $C_t$ value of the endogenous reference genes from the average $C_t$ value of the target gene. The $\Delta\Delta C_t$ value was then calculated by subtracting the $\Delta C_t$ value of the treated sample from the $\Delta C_t$ value of the untreated control.

Finally, the gene expression levels were calculated as $2^{-\Delta\Delta Ct}$ giving the final value that was normalised to the reference genes and relative to the control sample values of the studied genes. GenEx Enterprise software (MultiD Analyses, Göteborg, Sweden) for quantitative real-time PCR (qRT-PCR) expression profiling, was used to analyse and normalise the qRT-PCR data (*Kubista & Sindelka, 2007*).

## Phytochemical screening using GC/MS

A total of 5 mg of aqueous root extracts was re-dissolved in 5 mL of deionised water and filtered using polytetrafluoroethylene (PTFE) 0.2 $\mu$m pore size. With the aid of a vial, 1.5 mL filtrate was placed in the autosampler. Phytochemicals were screened using GC/MS -QP2010 ULTRA (Shimadzu Corp., Kyoto, Japan) with electron impact ionisation. The column used was RTX (fused silica) column with the dimension of length 30.0 $m_x$ thickness 0.25 mm, and diameter 0.25 $\mu$m (*Srivastava, Mukerjee & Verma, 2015*).

The GC/MS analysis was carried out under the following conditions; sample injection was performed using a 10 $\mu$L syringe with an injection volume of 1 $\mu$L. Helium gas was used as a carrier gas with a flow rate of 4.8 mL/min with a split ratio of 1:10. The temperature setting for the column was 40 °C, which was later increased at 15 min of the start-up by 10 °C per minute to 300 °C. This temperature was maintained for 30 min. The total run time/post run time was 48 min. The injector was set at room temperature, and detector voltage was 0.86 kV. The mass spectrometer scan ranges were from 28–500 amu at 1 spectra/sec, with ionising voltage and ionisation current of 70 eV (*Yuet et al., 2013*; *Srinivasa, Ammani & Rose, 2015*).

## Statistical analysis

The results were expressed as a mean $\pm$ standard error of the mean (SEM). One way analysis of variance (ANOVA) was employed for data comparison. GenEx statistical analysis was used for the gene expression, and analysis was performed using ANOVA followed by Tukey's post hoc test. A $p$-value $<0.05$ was considered indicative of a statistically significant difference.

# RESULTS

## Evaluation of the oestrogenic activity of *A. africanus*

The ethinyl estradiol (EE) treated group showed a significantly reduction in weight gain as compared to control for both with ORA ($p < 0.005$) and without ORA ($p < 0.005$). Figure 1 shows that the weight gains of groups treated with AEAA without ORA were 25.8, 40.5 and 57.6 g for the doses 50, 200 and 800 mg/kg, respectively. The differences were not significant as compared to control group (57.0 g). On the other hand, all AEAA treatment groups pre-treated with ORA showed significant ($p < 0.005$) reduction in weight gain versus control (Fig. 1). The weight gain reduction in the groups pre-treated with the ORA was in a dose-dependent manner, which could be due to a positive synergy between AEAA and ORA.

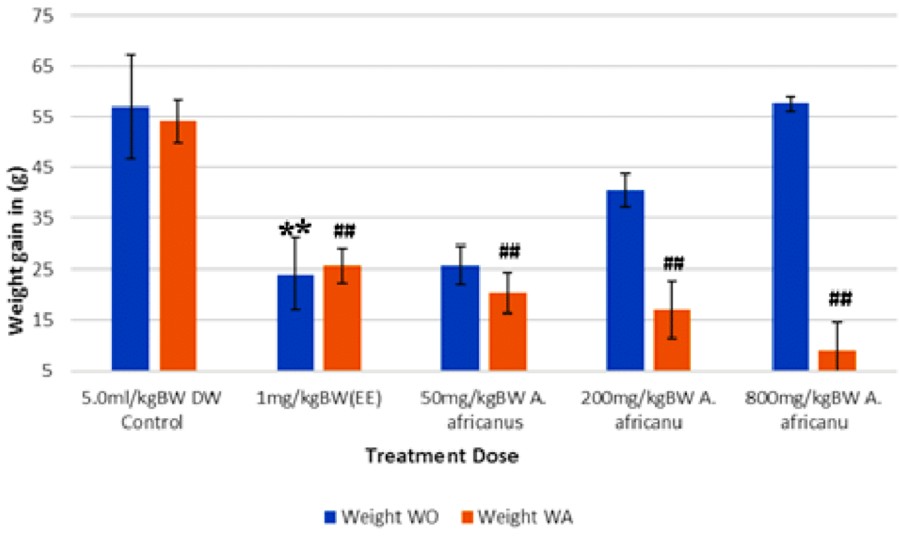

**Figure 1  Weight gain of rats treated with and without oestrogen receptor antagonist.** Values are expressed as mean ± SEM, ($n = 6$), \*\*$P < 0.005$ vs. control without antagonist, \#\#$P < 0.005$ vs. control with antagonist. WO, without antagonist; WA, with antagonist; EE, ethinyl estradiol.

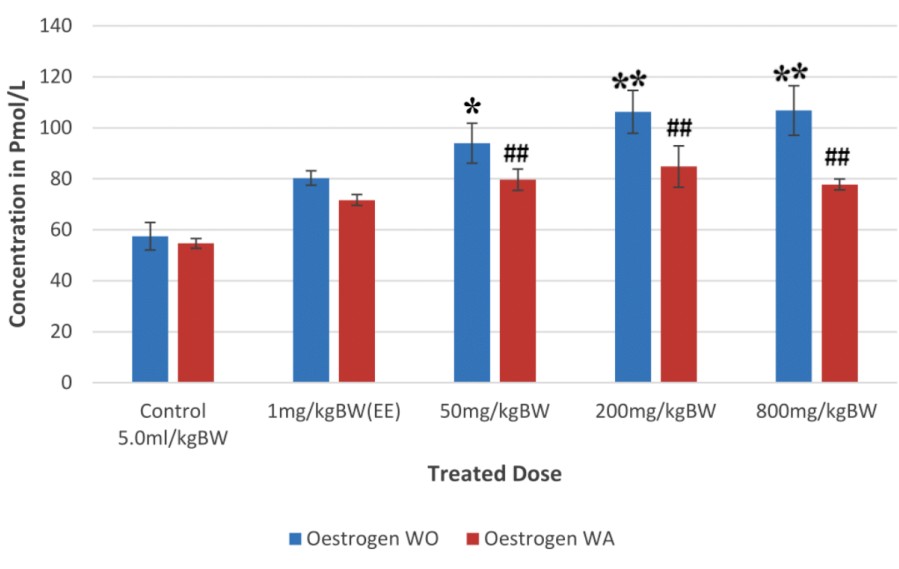

**Figure 2  Oestrogen levels in rats treated with and without the administration of oestrogen receptor antagonist.** Values are expressed as mean ± SEM, $n = 6$, \*$P < 0.05$, and \*\*$P < 0.005$ vs. control without antagonist, \#\#$P < 0.005$ vs. control with antagonist. WO, without antagonist; WA, with antagonist; EE, ethinyl estradiol.

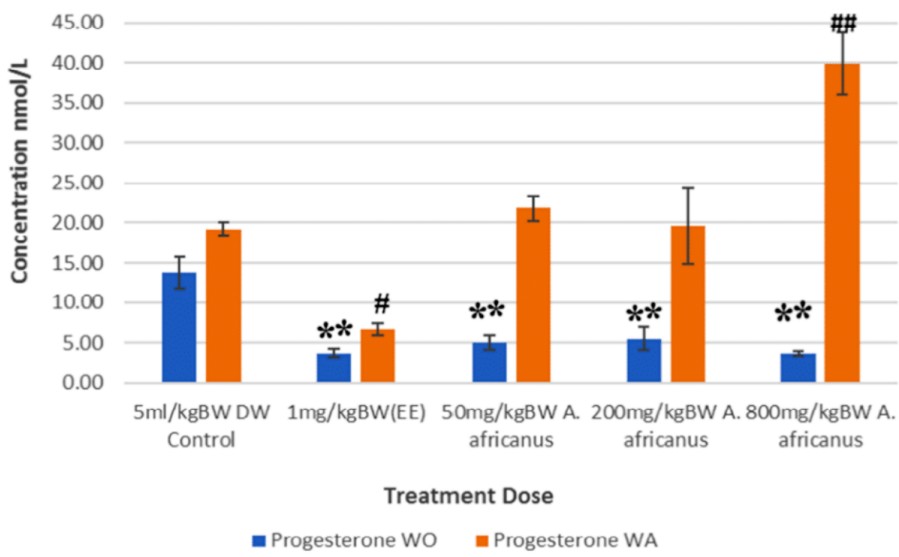

**Progesterone for rat treated with and Without the estrogen antagonist**

Figure 3 **Progesterone levels in rats treated with and without oestrogen receptor antagonist.** Values are expressed as mean $\pm$ SEM, ($n = 6$), $**P < 0.005$ vs. control without antagonist, $^{\#}P < 0.05$, and $^{\#\#}P < 0.005$ vs. control with antagonist. WO, without antagonist; WA, with antagonist; EE, ethinyl estradiol.

The concentration of oestrogen (Fig. 2) also depicted a dose-dependent increased pattern with a significant increase in the treated animals without antagonist as compared to control. The mean oestrogen levels of rats treated with AEAA in the absence of ORA were 94.0 ($p < 0.05$), 106.3 ($p < 0.005$) and 106.8 ($p < 0.005$) pmol/L for the doses 50, 200 and 800 mg/kg, respectively, which were significantly higher than control group (57.5 pmol/L). Administration of ORA prior to the treatment reduced the levels of oestrogen as compared to the same group without the antagonist. However, the level of oestrogen in all treated groups was significantly ($p < 0.005$) higher as compared to the control rats pre-treated with the antagonist.

The mean progesterone levels of rats treated with AEAA in the absence of ORA were 5.05, 5.53 and 3.68 nmol/L for the doses 50, 200 and 800 mg/kg, respectively, which were significantly ($p < 0.005$) lower than control group (13.75 nmol/L). However, with the administration of ORA, the concentration of progesterone increased significantly ($p < 0.005$) in 800 mg/kg *A. africanus* treated rats (Fig. 3). Figure 4 shows the result of luteinising hormone (LH) for treated and untreated SD rats and the results were statistically not significant between groups.

## Gene expression

Figures 5 and 6 show the genes expression levels of rats treated without and with ORA, respectively. Treated rats with 200 mg/kgBW *A. africanus* showed significant ($p < 0.005$) upregulation of ER α with 7.47 folds as compared to calibrator (Fig. 5). While for the

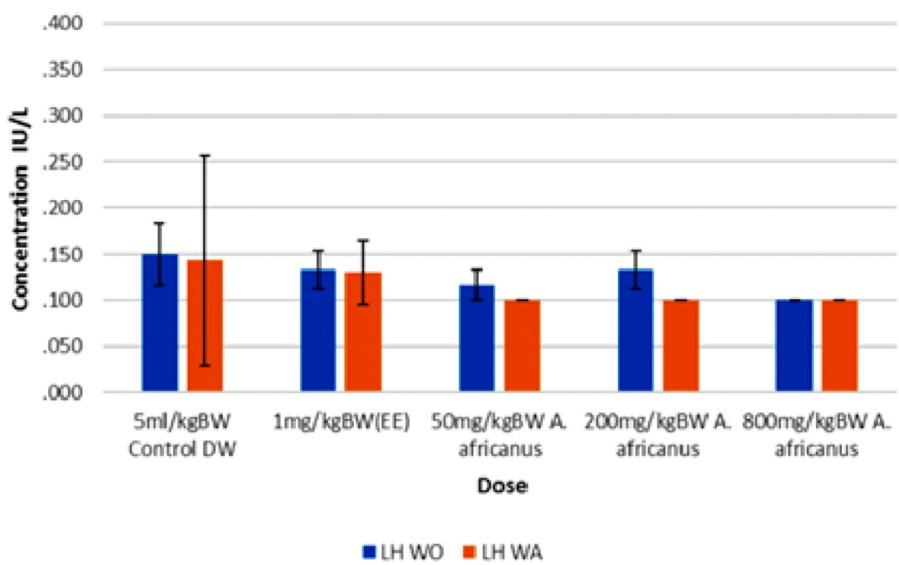

**Figure 4 Luteinizing Hormone (LH) levels in rats treated with and without oestrogen receptor antagonist.** Values are expressed as mean ± SEM, ($n = 6$). WO, without antagonist; WA, with antagonist; EE, ethinyl estradiol.

treated animals with ORA, the Calb3 of ethynyl estradiol-treated group was significantly ($p < 0.005$) overexpressed with 7.64 folds as compared to the calibrator (Fig. 6).

## GC/MS chromatogram analysis

GC/MS chromatogram of the aqueous root extract of *A. africanus* (Fig. 7) shows nine identifiable peaks indicating the presence of nine phytochemical compounds. Chromatograms and spectrums obtained were identified using National Institute of Standards and Technology Library (NIST). Mass Spectrum Search Interpreter version 11 database was used for possible chemical identification by comparing the spectrum and the mass to charge ratio (m/z) of the identified peaks. The details of the compounds are presented in Table 1. The findings revealed two phytoconstituents with steroidal nucleus namely stigmasterol and sarsasapogenin that are structurally related to hormone backbones. Furthermore, other non-steroidal compounds such as; 2(3H)-furanose, dihydro-3-hydroxy-4,4-dimethyl; pyrazinetetramethyl; 1,2-benzenedicarboxylic acid; 7,9-Die-tertt-butyl-1-oxospiro (4,5) deca-6,9-diene-2,8-dione; n-Hexadecanoic acid; butyl citrate; and 3-Dehydro-des-N-26-methyl-dihydro-pseudotomatidine were also detected. The mass spectrum of the identified phytochemicals of aqueous root extract of *A. africanus* are attached as supplementary materials and labelled as Figs. S1–S9.
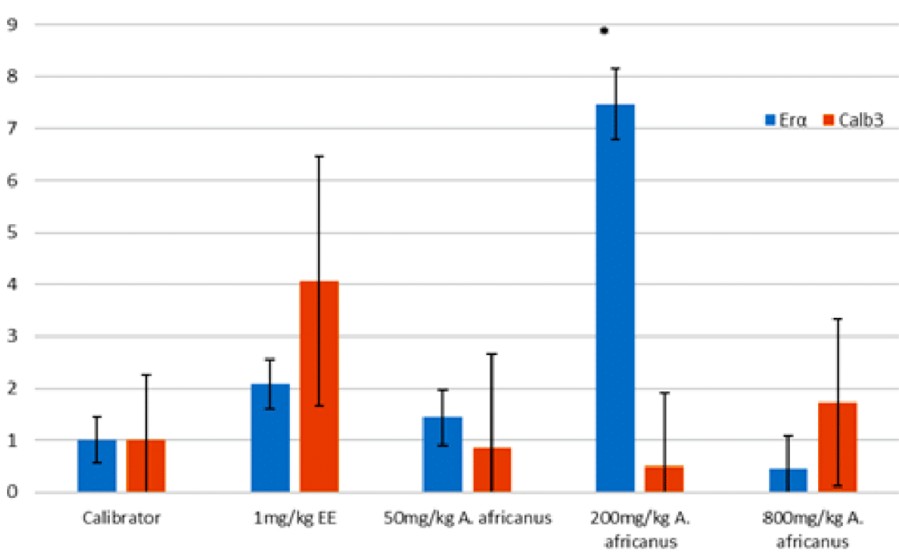

**Figure 5 Graphical representation of gene expression of rats treated without oestrogen receptor antagonist.** Values were expressed and presented as mean ± SEM, ($n = 4$), *$p < 0.005$ vs. calibrator without oestrogen receptor antagonist. EE, ethynyl estradiol; Erα, oestrogen receptor alpha gene and Calb3, calbindin 3 gene.

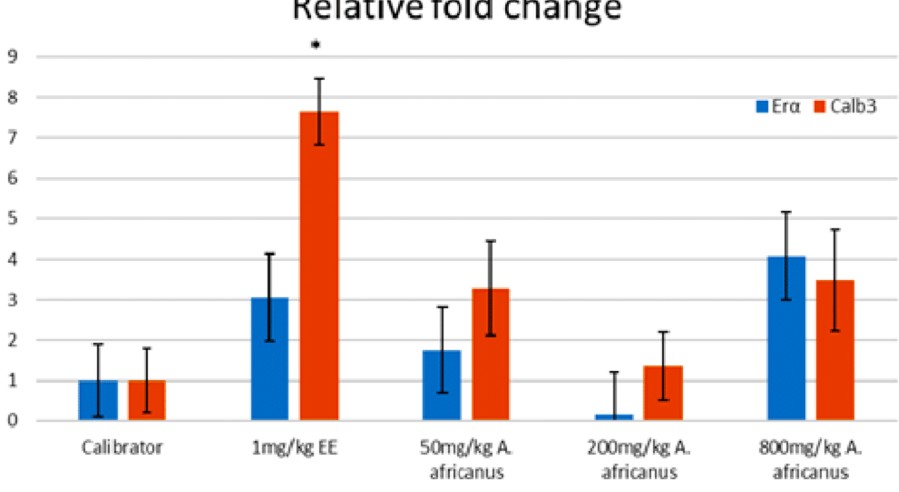

**Figure 6 Graphical representation of gene expression of rats treated with oestrogen receptor antagonist.** Values were expressed and presented as mean ± SEM, ($n = 4$), *$p < 0.005$ vs. calibrator with oestrogen receptor antagonist. EE, ethynyl estradiol; Erα, oestrogen receptor alpha gene and Calb3, calbindin 3 gene.

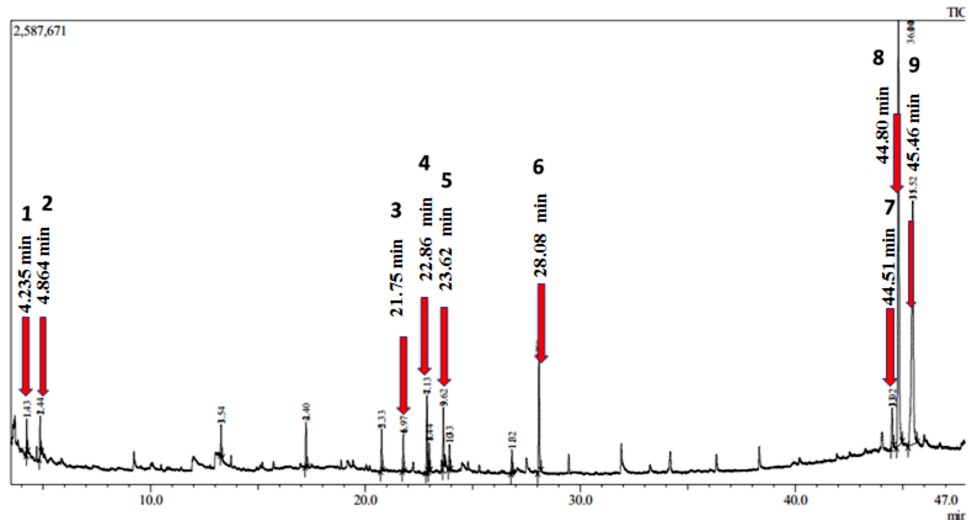

**Figure 7** Chromatogram obtained from GC/MS analysis of aqueous root extract of *A. africanus.*

**Table 1** Phytochemicals identified in *aqueous root extract of A. africanus* using GC/MS.

| Peak No | RT (min) | Area (%) | Height (%) | m/z | MW (g/mol) | CAS | Chemical formula | Tentative name |
|---|---|---|---|---|---|---|---|---|
| 1 | 4.235 | 4.88 | 7.29 | 71.10 | 130 | 79-50-5 | $C_6H_{10}O_3$ | 2(3H)-furanose,dihydro-3-hydroxy-4,4-dimethyl |
| 2 | 4.864 | 4.37 | 6.87 | 54.10 | 136 | 1124-11-4 | $C_8H_{12}N_2$ | Tetramethylpyrazine |
| 3 | 21.75 | 6.91 | 8.25 | 149.05 | 278 | 84-69-5 | $C_{16}H_{22}O_4$ | 1,2-Benzenedicarboxylic acid, bis(2-methylpropyl) |
| 4 | 22.86 | 2.97 | 3.85 | 57.15 | 276 | 82304-66-3 | $C_{17}H_{24}O_3$ | 7,9-Die-tert-butyl-1-oxospiro (4,5) deca-6,9-diene-2,8-dione |
| 5 | 23.62 | 2.95 | 3.40 | 43.15 | 256 | 57-10-3 | $C_{16}H_{32}O_2$ | n-Hexadecanoic acid |
| 6 | 28.08 | 12.09 | 16.01 | 185.10 | 360 | 77-94-1 | $C_{18}H_{32}O_7$ | Butyl citrate |
| 7 | 44.51 | 1.82 | 1.68 | 55.15 | 412 | 77-94-1 | $C_{29}H_{48}O$ | Stigmasterol[a] |
| 8 | 44.80 | 40.78 | 34.07 | 139.10 | 416 | 126-19-2 | $C_{27}H_{44}O_3$ | Sarsasapogenin[a] |
| 9 | 45.46 | 23.18 | 18.55 | 139.10 | 414 | 0-00-0 | $C_{28}H_{46}O_2$ | 3-Dehydro-des-N-26-methyl-dihydro-pseudotomatidine |

**Notes.**
[a]Steroidal saponins in the *aqueous root extract of A. africanus.*

## DISCUSSION

The oestrogenic activity of plants which blocks foetal angiogenesis was suggested to probably work through exerting anti-implantation (*Reese et al., 2001*; *Mukesh et al., 2006*). *Reese et al. (2001)* reported that oestrogen secretion is very important for proper implantation of the blastocyst with the uterine wall during early pregnancy. In our study, the AEAA demonstrated an increase in the level of oestrogen and a severe reduction in the progesterone in a dose-dependent manner. This activity is anti-progestin in action, and progesterone is required in preparation of the uteri for proper implantation of the blastocyst to the uterine wall through the angiogenesis process. This could be a possible mechanism through which AEAA exerts its anti-implantation activity.

Our study revealed that AEAA has antiprogestins, and in the absence of progesterone, pregnancy cannot be initiated or maintained. The main function of the corpus luteum

is secretion of progesterone hormone, which is important during the luteal phase of the menstrual cycle and for maintenance of normal pregnancy in mammals (*Donald, 1999*). We observed that concentrations of progesterone remained significantly low for all the treated groups ($p < 0.005$) when compared with the control group. Moreover, with the pre-administration of ORA, the level of progesterone increased significantly ($p < 0.005$) in 800 mg/kgBW of AEAA treated group. Furthermore, luteinising hormone levels in AEAA treated and untreated SD rats showed non-significant changes.

There are multiple mechanisms through which phytoestrogens exert its effect, including alterations of oestrogen receptor expression (*Andreana & Edward, 2003*). In our study, the AEAA at 200 mg/kgBW has significantly upregulated the expression of ERα in the rats without ORA and down-regulated in the presence of the antagonist. Previous studies reported that AEAA might be a selective oestrogen receptor modulator (*Ermias et al., 2014*). In our study, the result showed a selective pattern in terms of activities of AEAA on oestrogen receptors, which is an example of selective estrogen receptor modulator (SERM) and an ideal SERM would have antagonist activity on the uterus while having an agonist activity on other target tissues that profit from an oestrogen-like action (*Ilaria et al., 2014*).

Expression of Calb3 in the uterine of endometrium increased during the pregnancy and the estrus cycle (*Yohan et al., 2012*). Likewise, there were changes during gestation of calb3 in the uterus and placenta (*Catherine et al., 1989*). AEAA at a concentration of 200 mg/kgBW has down-regulated the expression of calb3 gene in the absence of the ORA, however this reduction was not statistically significant. On the other hand, Calb3 expression was upregulated after administration the ORA, but again the overexpression was not statistically significant. Oestrogen has been known to increase the expression of Calb3 during early pregnancy (*Yohan et al., 2012*). Our finding also showed that EE significantly increased the expression of Calb3 in rats without ORA, which is in line with the reported data by Yohan et al., who showed that the expression of Calb3 in the uterine endometrium was under the regulation of oestrogen during pregnancy (*Yohan et al., 2012*). The increment in the level of estradiol concentration was usually connected to the increase in the level of Calb3 (*Hong & Jeung, 2013*).

The relationship between the dose of a bioactive compound and the pharmacological response (dose–response curve) is not always appears as a linear curve. In our study the level of progesterone and gene expressions of ERα and Calb3 showed a non-monotonic dose–response curve, which is inconsistent with the typical curve of dose–response relationship. Such a dose response curve has been also reported with some endocrine disrupting compounds such as bisphenol A (*Vandenberg, 2014*). One of the proposed mechanisms resulting in such kind of curve is the disruption in hormonal regulation (*Vandenberg et al., 2012*; *Lagarde et al., 2015*; *Vandenberg, 2014*). This kind of non-linear dose–response curve could be explained by the low-dose effects (*Do et al., 2012*; *Vandenberg, 2014*; *Lagarde et al., 2015*). Hence, we postulate that some of the findings of our study suggest that the disruption on the hormonal regulation could have been compensated by negative feedback response.

Many phytochemicals isolated from plants are capable of initiating and modulating hormonal activities. A chemical isolated from the female part of *Humulus lupulus* "hops"

is an example of endocrine disruptor found in medicinal plant. The inflorescences parts of *H. lupulus* contains xanthohumol, which is converted by large intestine normal flora into 8-prenylnaringenin. The phytoestrogen 8-prenaylnaringenin is a bioactive compound of hops and one of the most potent phytoestrogens isolated until now has binding affinity to estrogen receptors and may effect on menstrual cycles and decreases the fertility rate (*Milligan et al., 2002*).

The investigation of *A. africanus* root extract shows the presence of polyphenols, phytosterol, saponins and tannins (*Yared, Mekonnen & Debella, 2012*). These phytochemicals have been implicated to have oestrogenic properties in some animal species (*Nwafor, Okwuasaba & Onoruvwe, 1998*). Findings from GC/MS analysis revealed the presence of phytoconstituents with a steroidal nucleus, namely stigmasterol and sarsasapogenin, which are structurally related to hormones backbone. This result supports the findings by Geremew who reported that plants might possess hormonal properties capable of modulating the reproductive function of the experimental rats (*Geremew, Yalemtsehay & Eyasu, 2006*). The presence of steroidal saponins has also been reported by Asfaw et al. which is in agreement with our study (*Asfaw et al., 1999*). The findings of this study confirm that *A. africanus* contains potential steroid-like compounds, which could explain the traditional uses of its root as pre-natal alternative medicine or contraceptive.

## CONCLUSION

The female rats treated with the root extract of *A. africanus* showed a significant increase in the production of oestrogen and reduced levels of progesterone, which may indicate that *A. africanus* root extract, could exert an abortifacient effect mainly by the oestrogenic and antiprogestin effects. The GC/MS analysis of the root extract of *A. africanus* showed the presence of steroidal saponins; stigmasterol and sarsasapogenin. These phytochemicals could explain the traditional medicinal uses of *A. africanus,* especially in birth-related applications. Further pharmacological studies should be carried out to investigate the efficacy of the plant, especially the antifertility activities of the steroidal saponins identified in this plant.

### Funding
This study was supported by the IPPP grant no. PG112-2014B from the University of Malaya, Malaysia and postgraduate scholarship from the Federal Polytechnic, Damaturu Nigeria (Tetfund). The funders had no role in study design, data collection and analysis, decision to publish, or preparation of the manuscript.

### Grant Disclosures
The following grant information was disclosed by the authors:
University of Malaya: PG112-2014B.
Federal Polytechnic, Damaturu Nigeria (Tetfund).

## Competing Interests

The authors declare there are no competing interests.

## Author Contributions

- Abubakar El-Ishaq performed the experiments, analyzed the data, prepared figures and/or tables, authored or reviewed drafts of the paper, approved the final draft.
- Mohammed A. Alshawsh analyzed the data, contributed reagents/materials/analysis tools, authored or reviewed drafts of the paper, approved the final draft.
- Zamri Bin Chik conceived and designed the experiments, contributed reagents/materials/analysis tools, authored or reviewed drafts of the paper, approved the final draft.

## Animal Ethics

The following information was supplied relating to ethical approvals (i.e., approving body and any reference numbers):

The ethics committee for animal experimentation, Faculty of Medicine, University of Malaya, Malaysia approved this study (Ethics Reference no.: 2015-180505/PHAR/AEI).

## Data Availability

The raw data are available as a Supplemental File.

## Supplemental Information

Supplemental information for this article can be found online at http://dx.doi.org/10.7717/peerj.7254#supplemental-information.

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
