# Peer review of "Evaluating the oestrogenic activities of aqueous root extract of Asparagus africanus Lam in female Sprague-Dawley rats and its phytochemical screening using Gas Chromatography-Mass Spectrometry (GC/MS)"

_PeerJ, doi:10.7717/peerj.7254_

## Round 0.1 · original submission · Major Revisions

All reviewers see value in your work but have some major concerns, which need to be addressed. Thank you for sharing your research.

Reviewer 1 ·

Basic reporting

1.1 English is clear and professional english.
1.2 References in the main text are numbered but the reference section is just alphabetical. Which makes a detailed review impossible.
1.3 Introduction: Add sentence for what specifically asparagus africanus is used for in traditional medicine?
1.5 I am lacking some more discussion and examples of other plants which are known to be endocrine disruptors and their possible use and interaction effects in medicine. E.g. Female hop (Lupuli flos) ploughers can have varying menstrual cycles and decreased fertility rates.
1.6 line 93: Weight of the extract after drying should be noted in the manuscript.
1.7 Line 254: “This dose …” Unclear about what dose the authors talk?
1.8 Line 257: Abbreviation “SERM” not introduced in the manuscript and meaning is unclear.
1.9 Figures are relevant, high quality, well labelled & described. Raw data is attached which helps with the review.

Experimental design

2.1 The primary research is within the scope of the journal.
2.2 Research question well defined, relevant & meaningful. It is stated how the research fills an identified knowledge gap.
2.3 Line 91: Has the extraction been performed under light or no light?
2.4 Otherwise experimental details and results are well described and acceptable for publication.

Validity of the findings

3.1 Several of the author conclusions are well stated but I am unable to link them to the supporting results. In general, the results give more questions than answers and I would like to explain and discuss that with the next points. In general, I m lacking to see dose-response curve like effects between the 50, 200 and 800 mg/kgBW groups for the various measured endpoints. It looks a little bit like that for the highest dose effects are different compared to the other two groups (downturn effect at too high doses). Dose-response effects have not been proper identified in the manuscript or discussed: The authors have not performed any form of trend test or dose-response models to confirm their conclusions (just ANOVA plus posthoc which tests nominal differences and not ordinal trends). This makes many of the conclusion’s speculations.
3.1.1 The interpretation of figure one is already very difficult.
In the group without ER receptor antagonist the weight gain is obviously the smallest in the lowest concentration groups comparable to EE, however, it increases dose dependent and is absent in the highest dose group compared to negative controls. Which gives the impression the effect decreases with the concentration of AEAA.
In the group with ER antagonist the weight gain is seemingly dose dependent decreasing compared to negative control. It looks like there is a positive synergy between AEAA and the ER antagonist.
3.1.2 Figure 2 and 3: The authors claim that “… the AEAA demonstrated an increase in the level of estrogen and a severe reduction in the progesterone in a dose-dependent manner.” (line 236f as well in the abstract). However, I can not see a dose dependent increase or decrease of any of the two in the group without ER receptor antagonist.
3.1.3 The authors implicitly claim that the phytochemicals they have identified are responsible for the effects they have observed. However, also other pharmacological interactions are possible and not discussed, mentioned or tested for. E.g. high doses of St John's-wort (Hypericum perforatum) induces CYP3A4 which is also responsible for the metabolization of several hormones and can lower the effectiveness of e.g. hormonal contraception therapies. The authors have not supported their speculative conclusions by additional experiments (e.g. repeating experiments with the identified pure phytochemicals.
3.2 For GC/MS how the metabolites are identified is unclear. Has the main fragment m/z just been confirmed to a reference spectrum from the literature or has the entire fragments spectra been compared to pure reference substances spiked in AEAA or pure? Some more explanations are needed here.

Additional comments

none

Reviewer 2 ·

Basic reporting

Overall, the article is well written, provides sufficient background information, figures and literature. However, the manuscript needs some editing. Throughout the manuscript spaces are missing between some words (see lines 20, 48, 62, 117,119, 131, 189, 280, 335, 339). Also in the literature section there should be either a free line or no free line between all the references. All references should be also numbered as cited in the manuscript. Some references miss complete publication details and only part of the cited papers have their doi's mentioned.

Experimental design

In general the experimental design is very good and the original primary research is within the aims and scope of the journal. The research questions are well defined, relevant and meaningful. The GC/MS section however needs some improvements. First of all, the GC/MS manufacturer should be mentioned to be consistent with other methodological parts of the manuscript. The reference 24 mentioned is a corporate website without detailed method information. Please provide a different reference paper. Furthermore, references 25 and 26 mentioned in the GC/MS method sections do not contain any GC/MS experimental data. Please replace with proper references. Please also mention whether this instrument is using electron impact ionization and that no derivatization of the sample was necessary. Also missing is the information how peak identification was performed. Where spectra libraries used or chemical standards? The sentence in lines 175/176 starting with "during sample analysis...." can be deleted. In line 174 please mention precisely at what time in the gradient the 10C/minute increase was started. Also in line 174 break up the long sentence after … per minute to 300C and then continue with "This temperature was maintained for 30 minutes.

Validity of the findings

The authors used various methods to characterize potential oestrogenic activities of an aqueous root extract of Asparagus africanus Lam in rats. The data generated by gene expression and hormonal analysis show a clear impact by the root extract. GC/MS analysis supports the finding by the identification of two steroidal saponins. The only problem I can see with this manuscript is that the entire findings are based on a single root sample hence missing statistically relevant data. It would have been great, if the authors would have performed GC/MS analysis of several more root samples to see whether similar amounts of steroidal saponins are present in individual root samples.

Reviewer 3 ·

Basic reporting

The basic language should be improved to ensure that an international audience can clearly understand your text. While there is a certificate that states that a native English-speaking colleague reviewed this manuscript, there are still many places where the language could be improved. Some examples of this include the abstract. Please mention the use of this plant as a contraceptive in the opening sentence, then mention that this use prompted research into what mechanisms may be involved. This would also improve the introduction.
Another example of where the language could be improved is found in lines 62-65. It is unclear what is being said. Is the intent to introduce the uterotrophic assay with something along the lines of “A model used to evaluate the effect of chemicals on estrogen receptors is the uterotrophic assay in which removal of the ovaries reduces endogenous estrogens, causing the uterus to shrink. Replacement of these hormones with external sources of estrogen causes a trophic response of the uterus. ”
Please correct the references. Citations in-line are numbered, but the reference section appears to be sorted alphabetically, making it difficult to determine whether the citations used are appropriate.

Experimental design

The experimental design is generally well reported, but there are some areas where the rationale for the design and the components used could be clarified. For example in line 103, the immature rat version of the uterotrophic assay was developed because immature rats already have a uterus that is highly responsive to exogenous estrogens. Why where the immature rats used for this study also ovariectomized? Another example from line 103 includes the diet, which ought to be specified, and a statement made about whether the diet is low in phytoestrogens or phytoestrogen-free, as diet has been identified in the OECD test guideline for uterotrophic assays as a potential confound in uterotrophic studies.
In line 107, please indicate why the extract was injected twice daily. Was there a concern that some of the polyphenols might have a short half life? Also please clarify whether 800 mg/kgBW was injected twice per day for a total dose of 1600 mg/kgBW/day, or 400 mg/kgBW was injected twice per day for a total dose of 800 mg/kgBW/day.
In all of the supplemental tables, please use color consistently. For example, both of the graphs in the BW WO and BW WA tabs use blue columns, then in the BW combined tab, BW WA is colored orange. Please make the columns in all of the WA tabs orange as well, to facilitate comparisons. It would also be helpful if the axes were consistent for all graphs for the same endpoint. For example, the Y axis for ER WO runs to 125, ER WA to 100, and ER combined to 140. This makes it difficult to compare the magnitude of response between the graphs.
In the supplemental tables, for the weight reports, please check that the correct values have been graphed and in the correct order. In the BW WO tab, the 50 mg/kg body weights are near controls, then there is a decrease at 200 mg/kg. In Figure A in the BW combined, the 50 mg/kg body weights are very low, then increase dose-dependently until at 800 mg/kg, the weights are close to controls.
On the BW Combined Figure A, please also indicate what ## refers to. Is it statistically significant compared to controls pre-treated with antagonist?
In the supplemental tables on the ER combined page, there are two version of the same graph, which reports estrogen levels for rats with and without antagonist pre-treatment. Please remove one graph. Please also remove the extraneous ##, **, and * in columns A and B.

In the Prg combined tab, please remove the extraneous ## and # in columns O-R.
Please be consistent about indicating statistical significance within this supplemental data. The progesterone WO is indicated in the graphs in the combined tab as being statistically significant for all treatments, but the graph in the PRG WO tab does not indicate any statistically significant effects.

Validity of the findings

The conclusions appear to be appropriate, given the findings.

Additional comments

Please see the annotated PDF for additional comments relating to data reporting and experimental design.

Annotated reviews are not available for download in order to protect the identity of reviewers who chose to remain anonymous.

---

## Round 0.2 · Minor Revisions

Both re-reviewers noted the improvement of the manuscript but some minor things need to be cleaned up. We are looking forward to receiving a final document.

Reviewer 1 ·

Basic reporting

I can confirm that all comments by all three reviewers have been sufficiently addressed.

No further comments.

Experimental design

I can confirm that all comments by all three reviewers have been sufficiently addressed.

No further comments.

Validity of the findings

I can confirm that all comments by all three reviewers have been sufficiently addressed, except my comment 3.1 (comment #8).
I would strongly suggest that the authors implement the first paragraph "The dose response curve...negative feedback response." of their response somewhere in the discussion section of the manuscript.
These comments are of vital importance to interpret the data.

No further comments.

Additional comments

The manuscript has significantly improved. It has reached a level to be ready for publication.

Reviewer 3 ·

Basic reporting

In general, the article has improved, with the exception of some inconsistent use of acronyms.

Experimental design

Reporting of the experimental design has improved and is generally well reported.

Validity of the findings

I have some concern about text in the discussion that does not appear to match the figures provided. I have made comments in the pdf to support this statement, but an example would be "AEAA at a concentration of 200 mg/kgBW down-regulated the expression of calb3 gene in the absence of the oestrogen receptor antagonist." This statement should refer to Figure 5, which presents gene expression in the absence of ORA. In figure 5, there are no calbindin bars with an asterisk to denote statistical significance. The authors need to double check that the statements made in the discussion match those in their figures and in the body text.

Annotated reviews are not available for download in order to protect the identity of reviewers who chose to remain anonymous.

---

## Round 0.3 · accepted · Accept

Thank you for the second revision. I hope you feel that the review process improved your article. Congratulations to this publication.